# FIITED: FINE-GRAINED EMBEDDING DIMENSION OPTIMIZATION DURING TRAINING FOR RECOMMENDER SYSTEMS

## ABSTRACT

Huge embedding tables in modern Deep Learning Recommender Models (DLRM) require prohibitively large memory during training and inference. Aiming to reduce the memory footprint of training, this paper proposes FIne-grained In-Training Embedding Dimension optimization (FIITED). Given the observation that embedding vectors are not equally important, FIITED adjusts the dimension of each individual embedding vector continuously during training, assigning longer dimensions to more important embeddings while adapting to dynamic changes in data. A novel embedding storage system based on virtually-hashed physically-indexed hash tables is designed to efficiently implement the embedding dimension adjustment and effectively enable memory saving. Experiments on two industry models show that FIITED is able to reduce the size of embeddings by more than 65% while maintaining the trained model's quality, saving significantly more memory than a state-of-the-art in-training embedding pruning method. On public click-through rate prediction datasets, FIITED is able to prune up to 93.75%-99.75% embeddings without significant accuracy loss.

## 1 INTRODUCTION

Huge embedding tables in modern Deep Learning Recommendation Models (DLRM) reach terabytes in size (Lian et al., 2022). Training DLRMs usually requires model parallelism (Ivchenko et al., 2022; Sethi et al., 2023), but even with embedding tables distributed over multiple compute nodes, memory still proves a scarce resource (Lian et al., 2022). Reducing the memory cost of embedding tables is crucial to enable efficient model training and deployment of DLRM and allow for sustainable model development.

The size of an embedding table is determined by the number of rows (i.e., hash size), the number of columns (i.e., embedding dimension), and the size of each value in the embedding. Cutting down any of them can lead to memory saving. This paper focuses on optimizing the embedding dimension, an open and active topic in recent recommender model research (Zheng et al., 2023). Despite recent works (Qu et al., 2022; Chen et al., 2021; Ginart et al., 2021; Liu et al., 2020; 2021; Yan et al., 2021b; Zhao et al., 2021; Zhaok et al., 2021; Xiao et al., 2022; Lyu et al., 2022; Kong et al., 2022; Yao et al., 2022), setting embedding dimensions in real-world applications still largely relies on empirical evidence, and the dimensions are usually uniform across different sparse features, which is presumably non-optimal.

Different from the normal practice, assigning suitable, *non-uniform* embedding dimensions is desirable for two reasons. First, typically, when a uniform dimension is used across sparse features, it is chosen to be large enough to accommodate the most information-rich features, which is an over-design to features with less information, leading to unnecessarily large memory consumption. Assigning custom embedding dimensions can naturally reduce the model size. Second, to achieve good model quality, the optimal embedding dimension is inherently non-uniform due to differences among features: exceedingly large dimensions can cause model overfitting, while too short dimensions are insufficient to express information contained in the sparse features.

Based on this insight, this paper proposes FIITED, a FIne-grained In-Training Embedding Dimension optimization method, where the dimension of each embedding vector is adjusted during train-

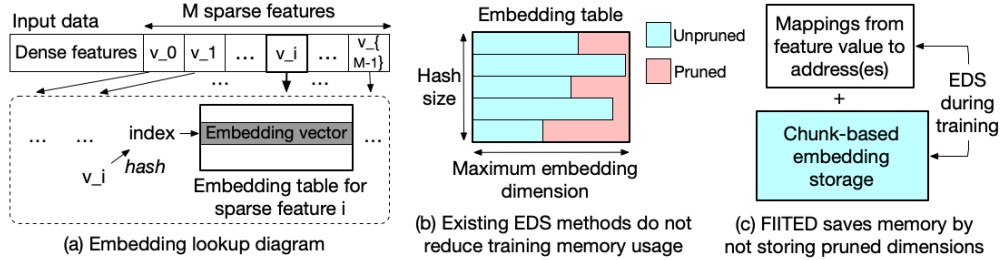

Figure 1: Existing Embedding Dimension Search (EDS) methods vs. FIITED.

ing. FIITED assigns more memory to more important embeddings and prunes the dimensions of less important embeddings to make better use of limited memory resources. Important embeddings are identified via importance scores computed during training which are based on embedding characteristics including access frequency and gradient norms. As a result, FIITED can reduce both the model size and the memory footprint during training to a desired amount.

Compared with previous Embedding Dimension Search (EDS) methods (Qu et al., 2022; Ginart et al., 2021; Chen et al., 2021; Liu et al., 2021; Zhao et al., 2021; Xiao et al., 2022; Lyu et al., 2022; Kong et al., 2022; Yao et al., 2022), the main advantage of FIITED, as shown in Figure 1, is enabling memory saving during training by not storing embeddings at the maximum length. Reducing the memory consumption of training is highly beneficial, because it enables training huge DLRMs on hardware devices with low on-device memory (e.g., GPUs) (Wang et al., 2022), lessens the need to spend extra time transferring embeddings from off-device memory-rich hardware (e.g., CPUs and SSDs) (Sethi et al., 2022; Zhao et al., 2020), and potentially frees up memory for more features to be added to the model, which can improve the model quality. However, it is non-trivial to realize, because naively pruning embedding dimensions will result in many tiny fragments of free memory which are hard to utilize. To tackle this challenge, we propose a novel chunk-based embedding storage system.

In addition to reducing the training memory footprint, FIITED brings three more advantages compared to most of the existing EDS methods. First, FIITED is fined-grained and adjusts the embedding dimension at the embedding vector level, while most EDS methods operate at the sparse feature level and sets a uniform dimension for all embeddings within the same sparse feature (Qu et al., 2022; Chen et al., 2021; Ginart et al., 2021). Fine-grained EDS is necessary because even within the same sparse feature, some embeddings may be more important than others and thus require different dimensions (Lai et al., 2023; Liu et al., 2020). Second, FIITED performs EDS *during training* and can take advantage of dynamic changes in data characteristics. Since the importance of an embedding vector often changes over time (Lai et al., 2023; Liu et al., 2020), it is natural to assume that their optimal dimension should vary accordingly. In-training EDS offers unique opportunities to adjust embedding dimensions over time during training, which is not allowed in pre-training or post-training EDS methods (Qu et al., 2022; Ginart et al., 2021; Chen et al., 2021; Zhao et al., 2021; Kong et al., 2022). Moreover, FIITED does not rely on any prior knowledge of the training data statistics but rather adapts to data characteristics during training, and therefore is better suited to application domains with fast changing data traits or when training data are not fully observable at the start of training, e.g., during online learning. Third, FIITED does not need any pre-training or re-training procedure, and thus has relatively low training time overhead.

In summary, this paper makes three main contributions:

- We propose FIITED, a novel fine-grained embedding dimension optimization method that adjusts the dimension of each embedding vector during training. FIITED directly cuts down training memory footprint by adopting a chunk-based embedding storage system design.

- The proposed method performs fine-grained EDS, adapts to changing data characteristics over time, and does not require pre-training, re-training or prior knowledge of training data traits. Thus, FIITED is more flexible, faster and easier to use than most of the existing EDS methods, while achieving higher reduction in the model size without hurting model quality.

- Experiments on two industry models show that FIITED can reduce a significant amount of embedding size during training (>65% and up to 50% for the two models, respectively) while maintaining the quality of the trained model. Compared to a state-of-the-art in-training embedding pruning method, AdaEmbed (Lai et al., 2023), FIITED is able to achieve higher pruning ratios without affecting model quality. On public datasets, FIITED is able to prune up to 93.75%-99.75% embeddings on three click-through datasets and 77.5%-81.25% on one classification dataset, without significant accuracy loss.

## 2 RELATED WORK

### 2.1 EMBEDDING DIMENSION SEARCH (EDS)

Existing EDS approaches can mainly be classified into three categories: (1) Pre-training, where embedding dimensions are decided before the actual training according to information extracted from the training dataset or a light-weight pre-training process (Qu et al., 2022; Liu et al., 2021; Ginart et al., 2021). Because the selected dimensions do not change during training, pre-training EDS misses opportunities to take advantage of changing data characteristics during training. Pre-training EDS also relies heavily on prior knowledge of the training data, which is not required by FIITED. (2) Post-training, where dimension pruning is performed after training. Additional information may be collected during training to aid pruning, and re-training is sometimes needed to boost the model performance (Chen et al., 2021; Zhao et al., 2021; Kong et al., 2022). Post-training EDS cannot reduce training memory footprint. (3) In-training, where the pruning decisions are made during training. Existing works (Liu et al., 2020; Yan et al., 2021b; Zhaok et al., 2021) add additional network structures on top of the original recommender model, e.g., new DNN layers, which help select embedding dimensions during training , but the majority of them still need to store embeddings at the maximum dimension at the time of training and thus do not reduce training memory usage. One exception is ESAPN (Liu et al., 2020), which stores embeddings at their current lengths during training by abandoning the old embeddings in memory whenever the embedding dimensions are increased. But without providing a system design, it is not clear how ESAPN can utilize the small pieces of free memory. ESAPN also does not provide control over how much memory is used during training, while FIITED can reduce the memory usage to an arbitrary desired amount. In addition, ESAPN trains one policy network per sparse feature and introduces significant training time overhead, while FIITED has negligible runtime overhead.

### 2.2 EMBEDDING COMPRESSION

To decrease the size of embedding tables, there are generally four ways: (1) reduce the size of each value in the table; (2) reduce the number of rows, i.e., the hash size; (3) reduce the number of columns, i.e., the embedding dimension; and (4) deconstruct the embedding table by replacing the traditional 2D array storage format with novel designs. (1) is commonly achieved by quantization (Zhang et al., 2018; Guan et al., 2019; Yang et al., 2020). (2) involves designing novel hashing methods to reduce the number of rows while maintaining the model quality (Zhang et al., 2020; Yan et al., 2021a; Desai et al., 2022). (3) has been discussed previously in Section 2.1 and includes EDS methods performed prior to, during or after training. Unlike existing EDS methods, FIITED is able to reduce the memory consumption during training without hurting the model accuracy or incurring significant performance overhead. For (4), prior research used several novel designs for embedding storage, including constructing embeddings from two separate tables (Pansare et al., 2022), via multi-layer embeddings (Ghaemmaghami et al., 2020), and by applying transformation matrices to a small set of anchor embeddings (Liang et al., 2020).

## 3 IN-TRAINING EMBEDDING DIMENSION OPTIMIZATION

In order to set appropriate embedding dimensions, one needs to capture the amount of information contained in embedding vectors, which varies across sparse features, across embeddings of the same sparse feature, and changes over time (Lai et al., 2023; Liu et al., 2020) . To tackle this challenge, we design an in-training embedding dimension optimization method that dynamically adjusts the dimension of each embedding vector during training.

## 3.1 FIITED OVERVIEW

Aiming to optimize embedding dimensions (i.e., dimension pruning), FIITED is built upon a state-of-the-art in-training embedding pruning method, AdaEmbed (Lai et al., 2023), which prunes entire embedding vectors during training (i.e., row pruning; each row in an embedding table stores one embedding vector). AdaEmbed assigns a row utility value to each embedding row based on row access frequency and gradient information, and the rows are pruned periodically by comparing the utilities against a pruning threshold. The threshold is decided by a pre-defined pruning ratio $p$. The current utility values are sorted in ascending order and the value at the $p$-th percentile is selected as the threshold. The value $p$ can be selected by the user and indicates the desired total size of embeddings. Rows with utilities below the threshold are pruned, while the rest remain.

To realize dimension optimization, we extend the row pruning method in AdaEmbed and divide every row in the embedding tables into K chunks, where each chunk is assigned its own pruning ratio. A utility metric is maintained during training for each chunk, and a total of K pruning thresholds are computed instead of 1, with 1 threshold per chunk. Chunks with utilities below the corresponding threshold are pruned. AdaEmbed becomes a special case of our dimension pruning method when K=1. As to how to decide the K pruning ratios, we provide two methods: (1) manual selection, and (2) dynamic generation at runtime. Details of the two methods will be explained in Section 3.2 and quantitative comparisons will be given in Section 4.

To further explain the idea, an example is illustrated in Figure 2. Each row in the diagram is an embedding vector with K embedding chunks, and each chunk has its own utility value. Given pruning ratios for the embedding chunks, a pruning threshold is computed for each chunk based on chunk utility values. For each embedding chunk, the pruning decision is made by comparing its utility against the pruning threshold. For example, in the first row, the first chunk's utility 1.3 is bigger than the threshold 0.15, so it is kept unpruned; the last chunk has utility 1.0 which is smaller than the threshold 1.4, so it is pruned. A previously pruned chunk can be brought back if its utility becomes larger than the threshold, and the chunk's embedding values will be re-initialized.

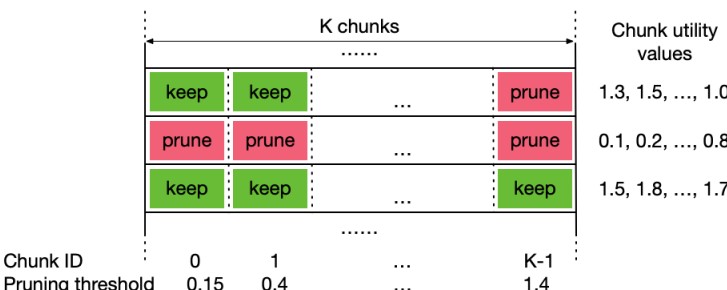

Figure 2: In-training embedding dimension pruning.

### 3.1.1 ZERO PADDING VS. LINEAR PROJECTION

The fine-grained dimension pruning in FIITED results in different embeddings having different lengths, which poses a challenge for the computation during training and inference. In DLRMs, dot product is usually computed among embedding vectors to capture their interaction, and it only works with embeddings of the same length. In previous works (Qu et al., 2022; Ginart et al., 2021), this problem is usually solved by inserting a Fully-Connected (FC) projection layer for each table to map embeddings of shortened lengths to the maximum length, so that any two embeddings will have the same length during dot product. But this strategy does not work for FIITED because even embeddings of the same sparse feature in the same table can have different lengths. Instead, we adopt zero padding to restore the embeddings to their original length, i.e., pruned chunks are treated as all zeros. If a fully pruned row is fetched during training, an all-zero embedding is returned.

Since computing dot products on zero-padded embeddings can cause information loss due to multiplications by zero, it may seem that padding zeros will restrict the model quality. But it is not an issue: the idea is that during training, the model can adjust itself and learn to accumulate in-

formation in the unpruned chunks. To corroborate our claim, we performed preliminary tests that compare FC layers with zero-padding during training of a Multi-Task-Multi-Label industry model. Prior to training, mixed embedding dimensions are assigned based on SVD analysis of a previously trained model, and they remain fixed during training. Model quality is measured by Normalized Entropy (NE) (He et al., 2014; Qin et al., 2020), with lower values indicating better models. Results (plotted in Figure 3) show that, compared to a baseline model with uniform embedding dimensions, zero padding only incurred a marginal NE loss (0.020%) in one of the three tasks while the other two tasks had tiny NE gains (-0.017%, -0.006%), indicating that zero-padding is indeed a feasible approach.

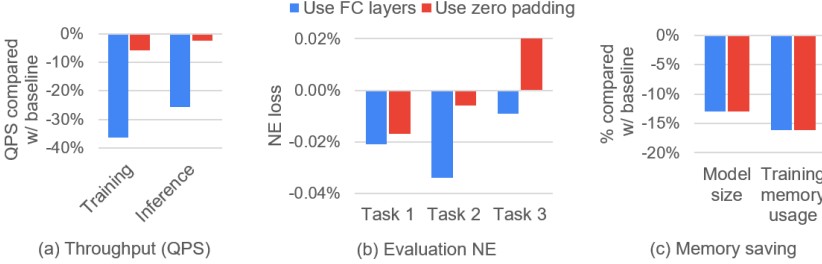

Figure 3: Zero padding vs. linear projection given the same per-feature embedding dimensions.

## 3.2 Dimension Optimization Algorithm

The dimension optimization procedure is detailed in Algorithm 1. Chunk utility values are updated in every training iteration. The utility values can simply record the running average of row access frequency, or it can be designed to incorporate more complex information, e.g., the L2 norm of gradients. Our preliminary results show that incorporating gradient information can lead to better model accuracy. Therefore, chunk utility in the $i$-th iteration is computed by $u(i) = \gamma u(i-1) + a(i)g(i)$, where $a(i)$ is the number of accesses to the chunk in iteration $i$, $g(i)$ is the L2 norm of the gradients computed for the chunk in iteration $i$, and $\gamma \in (0,1)$ is a decay parameter that reduces the influence of history utility values in order to capture dynamic changes in data characteristics.

Embedding pruning occurs every $T$ iterations. Empirically, $T$ is set to around 1 hour worth of training data. During pruning, pruning thresholds are computed by first sorting the current utility values and then selecting the value at the desired pruning ratio. Instead of sorting all the utility values which can take a long time, a small number of utility values are randomly sampled. After obtaining the thresholds, embedding chunks with utility below the thresholds are pruned while the rest are retained in the model.

### 3.2.1 Manual vs. adaptive per-chunk pruning ratios

Individual pruning ratios for each chunk can be selected either manually or adaptively according to data charcteristics. For manual selection, a pruning ratio $p_k$, $k = 0, 1, ..., K-1$, for each chunk is selected empirically before training. For adaptive selection, a global pruning ratio $p$ is decided by the user and indicates the average pruning ratio across all the chunks. Utility values in all chunks are sorted together to decide one global pruning threshold. Individual pruning ratios for different chunks may be bigger or smaller than $p$, and may change over time, depending on chunk utility values computed at runtime. We evaluate both approaches in the experiments (Section 4).

## 3.3 System design

Although fine-grained in-training dimension pruning has multiple benefits, it faces one crucial practical challenge that is to realize memory saving during training: a straightforward implementation will result in many small fragments of free memory. To this end, we propose a new Virtually Hashed Physically Indexed (VHPI) embedding table design adapted from AdaEmbed (Lai et al., 2023), as

---

**Algorithm 1** In-training embedding dimension optimization

---

   **Input:** pruning period $T$, sampling size $m$
   **for training batch ID** $i$ **do**
      Access embedding chunks, perform forward pass and backward pass
      Update embedding utility values
      **if** $i \% T = 0$ **then**
         Randomly sample $m$ utility values and sort them
         Compute pruning thresholds for embedding chunks
         **for every embedding chunk** $e_j$ **do**
            **if** utility value $u_j$ is below threshold and $e_j$ is not pruned **then**
               Set chunk $e_j$ to pruned (i.e., evict $e_j$)
            **end if**
         **end for**
         **for every embedding chunk** $e_j$ **do**
            **if** utility value $u_j$ is above threshold and $e_j$ is pruned **then**
               Set chunk $e_j$ to not pruned and initialize chunk $e_j$ (i.e., allocate $e_j$)
            **end if**
         **end for**
      **end if**
   **end for**

---

illustrated in Figure 4. The original system design in AdaEmbed becomes a special case where the number of chunks is 1.

**System Components** The design mainly consists of a hash table, an embedding table and a chunk address manager. (1) VHPI hash table. Each hash table entry contains addresses of K embedding chunks, K utility values and a bit mask that indicates whether the chunks have been pruned. Since K is usually chosen to be small, the size of an entry is also small, which allows the hash table to have a large number of entries and a low collision rate. (2) VHPI embedding table. The embedding table stores embedding vector chunks coming from all sparse features. The size of the embedding table is set to a desired amount decided by the user, i.e., the pruning rate multiplied by the size of embeddings in an unpruned model. (3) Chunk address manager. It manages free addresses in the embedding table and maintains a free address stack which is updated with newly available chunk locations whenever a chunk is evicted during pruning.

**Operations** (1) Embedding access. To fetch an embedding, the sparse feature ID and the feature value are together hashed to obtain the index of an entry in the VHPI hash table. Unpruned chunks are identified by checking the K-bit mask, and their addresses are obtained from the entry. Embedding chunks are then fetched from the embedding table according to the addresses. (2) Embedding eviction. When an embedding chunk needs to be evicted, its address is passed to the chunk address manager, who pushes the address into the free address stack, and the K-bit mask in the hash table is updated accordingly. (3) Embedding allocation. To allocate an embedding chunk, the chunk address manager fetches the next available chunk location, which is then stored in the hash table entry. Embeddings at the allocated location are initialized and the K-bit mask in the hash table is updated.

## 4 EXPERIMENTAL RESULTS

### 4.1 EXPERIMENTAL SETUP

**Baselines** We compare FIITED with the following baselines: (1) AdaEmbed (Lai et al., 2023), a state-of-the-art *in*-training embedding pruning method that prunes entire embedding rows during training, as described in Section 3.1; (2) ESAPN (Liu et al., 2020), a state-of-the-art *in*-training EDS method that selects embedding dimensions by training additional policy networks that decide to enlarge or shorten embedding dimensions during training; (3) Mixed Dimension (MD) embeddings (Ginart et al., 2021), a state-of-the-art *pre*-training EDS method that selects per-feature embedding dimensions prior to training based on access frequency; (4) last but not least, the original unpruned DLRM model with uniform embedding dimensions, which will be denoted simply as DLRM in the comparison.

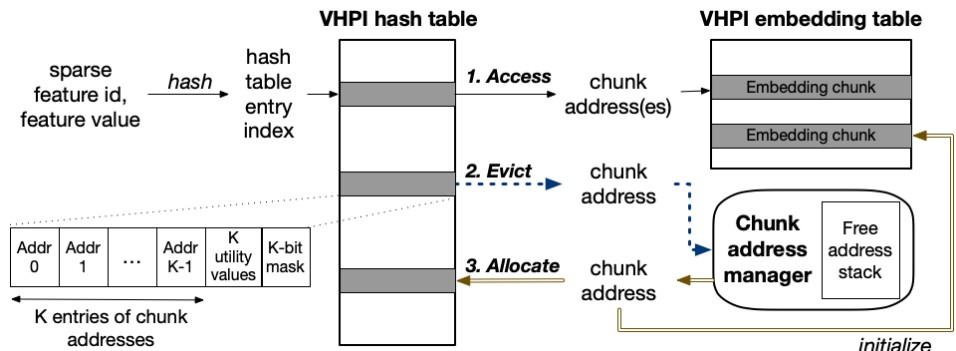

Figure 4: VHPI embedding table design that supports training memory saving in FIITED's chunk-based dimension pruning.

**Models and datasets** We evaluate FIITED on public models and datasets as well as private industry ones. (1) Public models and datasets: We adopt an open-source DLRM framework [1] to train a DLRM (Naumov et al., 2019) on Criteo Kaggle Display Advertising Challenge Dataset [2], Avazu Click-Through Rate Prediction Dataset [3] and Criteo Terabyte Dataset [4]. We also modified this framework to train another open-source DLRM [5] used by ESAPN (Liu et al., 2020) on the MovieLens-20M dataset [6]. On the MovieLens-20M dataset, following ESAPN, we converted the multi-classification problem to a binary classification problem by viewing 4-star and 5-star reviews as positive, others as negative. (2) Industry models and datasets: Two production models are used in the evaluation. For ease of experimentation, both models are shrunk to 1/4 size by reducing the hash size of all sparse features by 75%. The reduced models, named Model A and Model B, have around 180GB and 430GB sparse feature embeddings, respectively, and both contain hundreds of sparse features. They are evaluated on data generated by real-world applications. Ten consecutive days of data are used as the training set, and the following one day's data are used as the evaluation set.

**Implementation** For evaluation on the public models and datasets, we implement FIITED as well as AdaEmbed (Lai et al., 2023) on top of an open-source DLRM framework (Naumov et al., 2019) based on PyTorch. Utility update is executed in parallel with training to reduce runtime overhead. The implementation will be open-sourced upon acceptance of the manuscript. For evaluation on the industry models and datasets, a proof-of-concept prototype design of FIITED is implemented on the company's internal deep recommendation system code base. The prototype stores both pruned and unpruned embedding chunks, and executes pruning by setting the corresponding memory regions to zero.

**Training specifications** For evaluation on the public models and datasets, the experiments are run on 1 GPU, and we compare two methods to select per-chunk pruning ratios: manually and dynamically. To manually specify the pruning ratio for each embedding chunk, a linear function is fitted to satisfy the desired average pruning ratio. We used a default sparse embedding dimension of 32 on Criteo Kaggle and Avazu datasets, 64 on Criteo Terabyte and MovieLens-20M datasets. The number of chunks K is set to 2 on all datasets for FIITED. For evaluation on the industry models and datasets, the experiments are run on 4 compute nodes with 8 GPUs per node. Per chunk pruning ratios are manually chosen to roughly resemble a power law distribution, which was assumed by previous EDS work (Ginart et al., 2021). K varies between 4 and 8. Chunk utility is simply computed as a running average of access frequency in the experiments on industry models.

---

[1] https://github.com/facebookresearch/dlrm

[2] https://ailab.criteo.com/ressources

[3] https://www.kaggle.com/c/avazu-ctr-prediction

[4] https://labs.criteo.com/2013/12/download-terabyte-click-logs

[5] https://github.com/zgahhblhc/ESAPN

[6] https://grouplens.org/datasets/movielens/20m

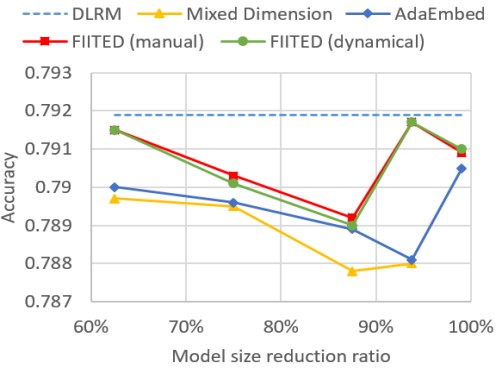

Figure 5: Validation accuracy under different pruning ratios (62.5%, 75%, 87.5%, 93.75%, 99%) on the Criteo Kaggle dataset. Mixed Dimension by design cannot reach 99% pruning rate.

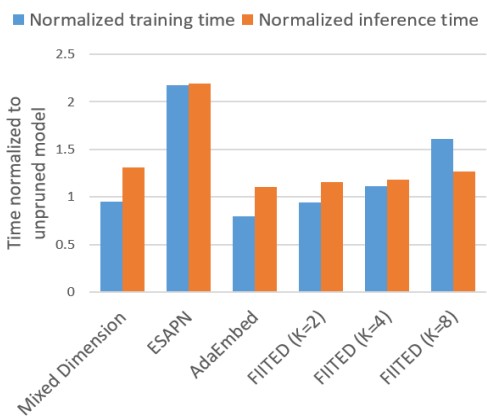

Figure 6: Average training/inference time per iteration of different models on Criteo Kaggle dataset (batch size=4096).

Table 1: Maximum embedding size reduction ratio on different datasets. ESAPN is only tested on the MovieLens-20M dataset, because its design only considers two types of embeddings, user embeddings and item embeddings, and does not apply to the other three datasets.

| Model | Criteo Kaggle | Avazu | Criteo Terabyte | MovieLens-20M |
|---|---|---|---|---|
| AdaEmbed | 98% | 92.5% | 99.5% | 72.5% |
| ESAPN | - | - | - | 51.9% |
| FIITED-Manual | **99%** | **95%** | **99.75%** | **81.25%** |
| FIITED-Dynamical | **99%** | 93.75% | **99.75%** | 77.5% |

## 4.2 RESULTS

### 4.2.1 TRAINING MEMORY SAVING

In this section, we evaluate how much memory can be saved by FIITED without affecting the model quality.

**Public models** For the public models, the criterion for model quality is prediction accuracy. According to Zhou et al. (2018) and Song et al. (2019), an accuracy loss larger than 0.1%-level is considered significant for click-through-rate prediction tasks. Table 1 shows the maximum pruning ratio of FIITED on different datasets, compared with AdaEmbed and ESAPN. On the three click-through-rate prediction datasets (i.e., Criteo Kaggle, Avazu and Criteo Terabyte), we reported the maximum pruning ratio without significant accuracy loss, while on the MovieLens-20M dataset we reported the maximum reduction ratio to obtain a comparable accuracy to ESAPN. On all four datasets, FIITED is able to prune more embeddings than the baselines while maintaining the model quality, and manual selection of pruning ratios performs a bit better than dynamical selection. Figure 5 shows the validation accuracy comparison of FIITED, AdaEmbed and Mixed Dimension on the Criteo Kaggle dataset using different pruning ratios . FIITED consistently achieves better accuracy than the baselines, and the two pruning ratio selection approaches (manual and dynamical) yield similar results. On the Criteo Terabyte dataset, FIITED is able to achieve a modest *0.1% accuracy gain* under a pruning rate of 93.75%.

**Production models** The criterion for model quality is Normalized Entropy (NE) (He et al., 2014; Qin et al., 2020) on the evaluation set; a lower NE indicates a better model. The effect of pruning is evaluated by NE loss, i.e., the percentage change in NE after model pruning compared to an unpruned model. A positive NE loss indicates worsened model quality, and an NE loss bigger than 0.02% is generally considered significant. The results are shown in Figure 7. For both Model A

and Model B, FIITED can achieve better NE than AdaEmbed given the same average pruning ratio. For Model A, without incurring significant NE loss (>0.02%), FIITED can prune 65%-75% of the embeddings, while AdaEmbed can only prune 50%-65%. For Model B, FIITED can prune up to 50% of the embeddings, while AdaEmbed can only prune 30%-40% of the embeddings without causing significant NE loss.

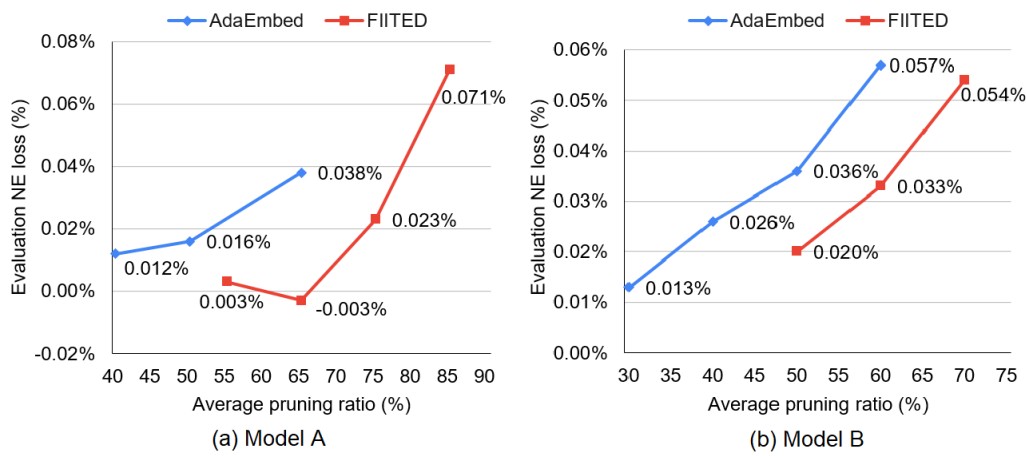

Figure 7: Model quality vs. pruning ratio. (a) Model A. (b) Model B.

### 4.2.2 OVERHEAD ANALYSIS

**Memory overhead** For a single-chunk FIITED model (i.e., K=1, same as AdaEmbed), the memory overhead can be estimated by $3/D$, where $D$ is the embedding dimension and 3 accounts for storing 1 chunk address, 1 utility value and 1 access frequency value per chunk. For K> 1, the memory overhead of FIITED grows in a linear scale and becomes $3K/D$. During inference, both the utility value and the frequency value can be discarded, resulting in a $K/D$ memory overhead.

**Performance overhead** Figure 6 shows the average training/inference time per iteration of MD embeddings, AdaEmbed, ESAPN and FIITED with $K = 2, 4, 8$, using Criteo Kaggle dataset with a batch size of 4096. The reported numbers are divided by the per-iteration time taken by an unpruned model for normalization purposes. As shown in the figure, during training, when $K = 2$, FIITED is able to outperform MD embeddings and the unpruned DLRM, while seeing a small overhead compared to AdaEmbed (i.e., FIITED with $K = 1$). The performance overhead grows linearly to the number of chunks. FIITED is significantly faster than ESAPN. During inference, the overhead of FIITED remains smaller than MD embeddings and ESAPN, and is slightly larger than AdaEmbed.

## 5 CONCLUSION

In this paper, we propose FIITED, an in-training embedding dimension optimization method that is able to directly cut down training memory footprint of DLRMs. Given a memory budget, FIITED can be plugged in directly into training without any need for prior knowledge of training data, pre-training or re-training. Embedding dimensions are adjusted during training in a fine-grained manner while changing data statistics are taken into consideration. Experiments on two industry models show that FIITED consistently achieves better NE than a state-of-the-art in-training embedding pruning method given the same average pruning ratio, and can prune much more than the baseline (65% vs. 50% for Model A, 50% vs. 30% for Model B) without affecting evaluation NE. On public datasets, FIITED is able to achieve an embedding size pruning ratio up to 93.75%-99.75% on three click-through-rate datasets and 77.5%-81.25% on one classification dataset, without significant accuracy loss.

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
