# OpenReview forum: "FIITED: Fine-grained embedding dimension optimization during training for recommender systems"
_ICLR.cc/2024/Conference — Submitted to ICLR 2024_

### Official Review · Reviewer_Rag8 · 2023-10-30

**Soundness:** 2 fair
**Presentation:** 1 poor
**Contribution:** 1 poor
**Rating:** 3
**Confidence:** 4

**Summary:**

This paper introduces a method called FIne-grained In-Training Embedding Dimension optimization (FIITED) to address the memory-intensive nature of modern Deep Learning Recommender Models (DLRM). These models have large embedding tables that demand a significant amount of memory during both training and inference. The core observation driving the research is that not all embedding vectors carry equal importance. Thus, FIITED dynamically adjusts the dimension of individual embedding vectors during training. It allocates longer dimensions to more critical embeddings and reduces dimensions for less significant ones, making optimal use of available memory. This optimization is based on the computed importance scores of embeddings, considering factors such as access frequency and gradient norms.

**Strengths:**

1.	The authors have shown an earnest effort to address the challenge of model compression, which remains a relevant topic in the field. Their intent to compress embedding dimensions can be useful in certain edge cases or specific scenarios.
2.	Despite similarities with existing methods, the introduction of a "Virtually Hashed Physically Indexed (VHPI) embedding table design" suggests the authors' endeavor to refine existing approaches.

**Weaknesses:**

Novelty and Contributions: The paper's novelty appears to be incremental and may not align with the standards of a top-tier conference. The area of embedding dimension search (EDS) has been previously explored. While the paper distinguishes itself by emphasizing feature value-level search, other works like AutoEmb [1], ESAPN [2], and RULE [3] have already pursued this direction. For the claimed novelty in searching during training, in-training searches using NAS-based optimization have been explored by DNIS [4], AutoEmb [1], and AutoDim [5]. The proposed Virtually Hashed Physically Indexed (VHPI) embedding table design seems reminiscent of what's presented in AdaEmbed. The only noticeable difference is the application of varying selection thresholds for different column groups in an embedding table.

Clarity and Motivation: The paper's motivation needs to be more explicitly articulated. The primary concern is the rationale behind the compression of embedding dimensions for standard recommender systems. When general recommendation models typically deploy on powerful cloud servers, the necessity to compromise model accuracy for size reduction seems unclear. The utility of adaptive embedding size search would be more apparent if it could enhance model performance by determining appropriate embedding sizes for different feature values. As it stands, while there's a notable compression rate, there's also a concomitant performance drop. Given the abundant memory and computational capacity of cloud servers, it's debatable whether such compression is essential or practical.

Quality of Figures: It's observed that the jpeg figure in the paper is somewhat blurred. It's recommended that authors use clearer formats like PDF to enhance the visual clarity of the figure.

Baseline Comparisons: The selection of baselines for comparison appears limited. Other potential methods such as DNIS [4], PEP [6], AMTL [7], and AutoDim [5] could have been considered or at least discussed in terms of baseline selection. A clear criterion for selection would lend more weight to the experimental results. Merely mentioning ESAPN as the state-of-the-art may not suffice.

Technical Details: There are missing technical details that might be crucial for a thorough understanding and replication of the method. The division of the embedding table into m*k embedding blocks is mentioned, but the determination of 'm' and the grouping strategy for different feature values are not elaborated upon. A comprehensive study or discussion on the hyperparameters, especially concerning 'm' and 'k', would be valuable. As of now, only Figure 6 sheds light on the varying training times with 'k'.

References:
[1] Manas R. Joglekar, Cong Li, Mei Chen, Taibai Xu, Xiaoming Wang, Jay K. Adams, Pranav Khaitan, Jiahui Liu, and Quoc V. Le. 2020. Neural Input Search for Large Scale Recommendation Models. In Proceedings of the 26th ACM SIGKDD International Conference on Knowledge Discovery & Data Mining (Virtual Event, CA, USA) (KDD ’20)

[2] Haochen Liu, Xiangyu Zhao, Chong Wang, Xiaobing Liu, and Jiliang Tang. 2020. Automated Embedding Size Search in Deep Recommender Systems. In Proceedings of the 43rd International ACM SIGIR Conference on Research and Development in Information Retrieval (Virtual Event, China) (SIGIR ’20).

[3] Tong Chen, Hongzhi Yin, Yujia Zheng, Zi Huang, Yang Wang, and Meng Wang. 2021. Learning Elastic Embeddings for Customizing On-Device Recommenders. In Proceedings of the 27th ACM SIGKDD Conference on Knowledge Discovery & Data Mining (Virtual Event, Singapore) (KDD ’21)

[4] Weiyu Cheng, Yanyan Shen, and Linpeng Huang. 2020. Differentiable neural input search for recommender systems. arXiv preprint arXiv:2006.04466 (2020).

[5] Xiangyu Zhao, Haochen Liu, Hui Liu, Jiliang Tang, Weiwei Guo, Jun Shi, Sida Wang, Huiji Gao, and Bo Long. 2021. AutoDim: Field-Aware Embedding Dimension Searchin Recommender Systems. In Proceedings of the Web Conference 2021 (Ljubljana, Slovenia) (WWW ’21).

[6] Siyi Liu, Chen Gao, Yihong Chen, Depeng Jin, and Yong Li. 2021. Learnable Embedding Sizes for Recommender Systems. arXiv preprint arXiv:2101.07577 (2021).

[7] Bencheng Yan, Pengjie Wang, Kai Zhang, Wei Lin, Kuang-Chih Lee, Jian Xu, and Bo Zheng. 2021. Learning Effective and Efficient Embedding via an Adaptively-Masked Twins-based Layer. In Proceedings of the 30th ACM International Conference on Information & Knowledge Management (CIKM '21).

**Questions:**

1.	Could you provide more insights into how the embedding table is divided into m*k blocks? Specifically, how is the value of 'm' determined, and what strategies are employed for grouping different feature values?
2.	On what grounds were the baselines selected for comparison? Why were certain other potential methods excluded from the baseline?

---

> ### Author Response · Authors · 2023-11-15
> **Response to questions**
>
> Thank you for your review!
>
> **Q1**. Could you provide more insights into how the embedding table is divided into m*k blocks? Specifically, how is the value of 'm' determined, and what strategies are employed for grouping different feature values?
>
> The embedding table is not divided into blocks. Every row in the embedding table (i.e., each embedding vector) is divided into K equal-sized chunks. For example, if the default embedding dimension is 128 and K=4, then the 4 chunks will be [0, 32), [32, 64), [64, 96) and [96, 128). Currently there is no grouping of feature values. Grouping was used in AdaEmbed to handle cases where there are different default embedding dimensions (e.g., one sparse feature may have a default dimension of 128 while another has 64). But this is not necessary for us, since in our experiments, there is only 1 default embedding size.
>
> I re-read the submission and could not find the notion of blocks. The notation m was used in Algorithm 1 as the sample size when sorting the utility values. Please let us know where "m*k blocks" was mentioned so that we can improve our wording and presentation.
>
> **Q2**. On what grounds were the baselines selected for comparison? Why were certain other potential methods excluded from the baseline?
>
> Because a main goal of FIITED is to save memory during training, we have selected baselines that can do the same. AdaEmbed is one such example where entire embedding rows are pruned during training. ESAPN [2] is another example where embedding dimensions are adjusted during training, and it claimed that memory saving can be achieved by always storing embeddings at their current lengths, although no system design was provided to show how the fragmented free memory can be utilized. Nevertheless, it was included in the comparison.
>
> Other potential methods are discussed in Section 2, including [3,5,6,7]. It will be interesting to see whether FIITED can reduce the model size more than these methods, but a crucial reason why they are not included in the baselines is that none of them [1,3-7] can actually save memory during training, and many of them increase the memory usage in training by introducing new structures to the recommender model. Although PEP [6] reduces memory usage in the retraining step, the pruning step still needs to store the entire unpruned embeddings as well as additional parameters used for pruning while training the model.
>
> We will add the discussion of baseline selection and include citations [1,4] into the manuscript.
>
> Please let us know if you have any questions or comments! Thank you.

---

> ### Author Response · Authors · 2023-11-19
> **Clarification of motivation**
>
> Although recommender models are usually trained on powerful cloud servers, as the model size keeps increasing, memory still tends to be the bottleneck. Also, as training machine learning models on edge devices becomes more and more popular nowadays due to privacy concerns etc., memory resources may be fairly scarse in some use cases.
>
> Model compression during training is a solution to this problem, and it does not necessarily lead to model quality deterioration. As briefly mentioned in Section 4.2.1, FIITED managed to achieve a 0.1% accuracy gain on the Criteo Terabyte dataset under a pruning rate of 93.75%. In other cases, FIITED can achieve superior compression rates while maintaining the model quality. Such results suggest that FIITED is able to find suitable dimensions for the embeddings while saving a considerable amount of training resources.

---

### Official Review · Reviewer_pdDD · 2023-10-31

**Soundness:** 3 good
**Presentation:** 3 good
**Contribution:** 2 fair
**Rating:** 3
**Confidence:** 2

**Summary:**

The research problem addressed in this paper is the challenge of optimizing embedding dimensions during training for recommender systems. The authors note that while embedding pruning methods have been proposed to address this issue, they often result in suboptimal performance due to the loss of information. To overcome this limitation, the authors propose a novel approach called Fine-grained In-Training Embedding Dimension Tuning (FIITED), which adjusts the dimension of each individual embedding vector continuously during training. This approach is designed to achieve significant memory savings without sacrificing model quality.

The authors note that previous work has explored various methods for embedding pruning, including manual pruning, adaptive pruning, and dynamic pruning. However, these methods have limitations, such as requiring manual tuning, being computationally expensive, or resulting in suboptimal performance. The authors argue that their proposed approach overcomes these limitations by allowing for fine-grained tuning of embedding dimensions during training.

The authors' contributions in this study include proposing the FIITED approach, which allows for fine-grained in-training embedding dimension tuning, and demonstrating its effectiveness through experiments on industry models and public click-through rate prediction datasets. The authors also propose a new Virtually Hashed Physically Indexed (VHPI) embedding table design adapted from AdaEmbed to address the memory fragmentation issue that arises from fine-grained pruning.

**Strengths:**

1. The authors validate the efficacy of FIITED through rigorous experimentation on both proprietary industry models and publicly available click-through rate prediction datasets. Specifically, they demonstrate that FIITED can reduce embedding size during training by more than 65% while maintaining the quality of the trained model. Compared to a state-of-the-art in-training embedding pruning method, AdaEmbed, FIITED is able to achieve higher pruning ratios without affecting model quality. On public datasets, FIITED is able to prune up to 93.75%-99.75% embeddings on three click-through datasets and 77.5%-81.25% on one classification dataset, without significant accuracy loss.

2. In addition to the FIITED methodology, the authors introduce an innovative embedding table design, termed Virtually Hashed Physically Indexed (VHPI), which is a modification of AdaEmbed. This design effectively mitigates the issue of memory fragmentation, a challenge commonly associated with fine-grained pruning. The authors demonstrate that VHPI can support training memory saving in FIITED's chunk-based dimension pruning.

**Weaknesses:**

1. The paper could benefit from a more detailed comparison with existing methods for embedding dimension optimization. While the authors briefly mention existing approaches such as EDS, they do not provide a comprehensive comparison with these methods. A more detailed comparison could help readers better understand the strengths and weaknesses of FIITED in relation to existing approaches.

2. The paper could provide more details on the implementation of FIITED and VHPI. Specifically, the authors could provide more information on the computational complexity of their approach and how it scales with the size of the dataset and the number of embedding dimensions. Additionally, the authors could provide more details on the implementation of VHPI, including how it mitigates the issue of memory fragmentation and how it compares to other embedding table designs.

3. The paper could benefit from more extensive experimentation on a wider range of datasets. While the authors demonstrate the efficacy of FIITED on both proprietary industry models and publicly available click-through rate prediction datasets, it would be useful to see how the approach performs on other types of datasets. Additionally, the authors could provide more details on the hyperparameters used in their experiments, such as the learning rate and batch size, and how they were selected.

**Questions:**

1. Can you provide more details on the computational complexity of your approach and how it scales with the size of the dataset and the number of embedding dimensions?

2. In Section 3.2.1, you mention that individual pruning ratios for different chunks may be bigger or smaller than the global pruning ratio, and may change over time, depending on chunk utility values computed at runtime. Can you provide more details on how these utility values are computed and how they are used to determine the pruning ratios for each chunk?

3. In Section 3.3, you introduce the Virtually Hashed Physically Indexed (VHPI) embedding table design, which is a modification of AdaEmbed. Can you provide more details on how VHPI mitigates the issue of memory fragmentation and how it compares to other embedding table designs?

4. In Section 4.2, you evaluate the performance of FIITED on four datasets, including three click-through rate prediction datasets and one classification dataset. Can you provide more details on the hyperparameters used in your experiments, such as the learning rate and batch size, and how they were selected?

5. In Section 4.3, you compare the performance of FIITED with that of AdaEmbed, a state-of-the-art in-training embedding pruning method. Can you provide more details on how AdaEmbed works and how it differs from FIITED?

6. In Section 5, you discuss the limitations of your approach, including the trade-offs between memory savings and model quality, and the limitations in terms of scalability and computational complexity. Can you provide more details on how these limitations might impact the applicability of your approach to large-scale datasets?

---

> ### Author Response · Authors · 2023-11-16
> **Response to questions - Q1**
>
> Thank you for your review!
>
> **Q1**. Can you provide more details on the computational complexity of your approach and how it scales with the size of the dataset and the number of embedding dimensions?
>
> For a dataset of size N with a total of M unique sparse feature values (i.e., M embeddings are needed for the model) and for a default embedding dimension of D divided into K chunks, the amount of extra computation introduced by FIITED is O(MKD) for maintaining utility values and O(M(K+D)) for pruning in 1 training iteration. If we consider a training epoch, the complexity becomes O(NMKD) for maintaining utility values and O(NM(K+D)) for pruning, because the number of training iterations in a training epoch is proportionate to the dataset size. The details are given below. But in reality, maintaining utility values is performed in parallel to training and in most cases does not increase training time. Pruning can indeed increase training time, but decreased computation and communication in training due to more sparsity in the embeddings can help decrease training time, so the performance overhead of FIITED stays relatively small.
>
> (1) Computation complexity of maintaining utility values
>
> When maintaining utility values, for every chunk a new utility value is computed in every iteration as follows (as mentioned in Section 3.2):
>
> u(i) = γu(i − 1) + a(i)g(i)
>
> where a(i) is the number of accesses to the chunk in iteration i, g(i) is the L2 norm of the gradients computed for the chunk in iteration i, u(i − 1) is the history utility value of the chunk computed in the previous iteration, and γ ∈ (0, 1) is a decay parameter that reduces the influence of the history utility value.
>
> Keeping track of a(i) takes extra computation, which is proportionate to batch_size * #embeddings_per_training_instance if we consider all chunks altogether. If we treat batch size as a constant, then this is roughly O(M) for all chunks combined.
>
> For every chunk, computing g(i) from gradients obtained in training is O(D), since each dimension has its own gradient value, Therefore, computing u(i) given a(i) is O(D), since the rest of the computation is constant time. For all chunks combined, the total computation is O(MKD), since g(i) needs to be computed for all M*K chunks.
>
> After g(i) is obtained for all chunks, a normalization process is performed to ensure utility values in different sparse features fall in a similar value range. For each sparse feature, a small constant number of utility values are sampled and then sorted to estimate the value at 95th percentile within the feature. Then every utility value within the sparse feature is divided by this 95th percentile value for normalization. This step takes O(MK) time.
>
> Therefore, for maintaining utility values in a training iteration, the time complexity is O(M) + O(MKD) + O(MK) = O(MKD).
>
> (2) Computation complexity of pruning
>
> As for pruning, first, a relatively small number of utility values are sampled to compute the pruning threshold. Since the sample size m is much smaller than M, it can be seen as a constant. Therefore, sorting the sampled values roughly takes constant time.
>
> Once the thresholds are obtained, for each chunk, its utility is compared to the threshold to decide whether it will be pruned. This step takes O(MK) time, because there are MK chunks in total.
>
> Once the pruning / un-pruning decisions are made, previously available chunks may be pruned, and previously pruned chunks may be brought back into the VHPI hash table. The complexity of pruning a chunk is O(1), because we only need to set the valid bit to 0 and send its address to the free address stack. The complexity of bringing a chunk back is O(D/K), because we need to set the valid bit to 1, get its address from the free address stack and initialize the memory region to all zeros. Initializing takes O(D/K) time, because there are D/K values to initialize. Because pruning / un-pruning can at most happen MK times, the total time complexity of performing pruning is O(MK * D/K) = O(MD) in the worst case.
>
> In total, pruning (i.e., getting thresholds, making pruning decisions, and the actual pruning action) is O(1) + O(MK) + O(MD) = O(M(K+D)), in 1 training iteration.
>
> (3) Conclusion
>
> In conclusion, maintaining utility values takes O(MKD) time and pruning takes O(M(K+D)) time in 1 training iteration. For 1 training epoch, maintaining utility values takes O(NMKD) time and pruning takes O(NM(K+D)) time.
>
> | Notation | Meaning |
> |---|---|
> | N | Dataset size |
> | M | Number of unique embeddings |
> | D | Embedding dimension |
> | K | Number of chunks per embedding |
>
> | Time complexity | for 1 training iteration | for 1 training epoch | Comments |
> |---|---|---|---|
> | Maintain utility values | O(MKD) | O(NMKD) | happens every training iteration; can be executed in parallel to training |
> | Periodic pruning | O(M(K+D)) | O(NM(K+D)) | happens infrequently (every few hundreds of training iterations) |

---

> ### Author Response · Authors · 2023-11-16
> **Response to questions - Q2**
>
> **Q2**. In Section 3.2.1, you mention that individual pruning ratios for different chunks may be bigger or smaller than the global pruning ratio, and may change over time, depending on chunk utility values computed at runtime. Can you provide more details on how these utility values are computed and how they are used to determine the pruning ratios for each chunk?
>
> As mentioned in Section 3.2 and also in Q1, the utility value of a chunk in the i-th iteration is computed as follows:
>
> u(i) = γu(i − 1) + a(i)g(i)
>
> where a(i) is the number of accesses to the chunk in iteration i, g(i) is the L2 norm of the gradients computed for the chunk in iteration i, u(i − 1) is the history utility value of the chunk computed in the previous iteration, and γ ∈ (0, 1) is a decay parameter that reduces the influence of the history utility value. The initial utility value is 0 at the start of training for every chunk. The utility value is updated in this way in every iteration for every chunk. More details for how utility values are computed are also included in Q1.
>
> As for deciding the pruning threshold, there are two ways. One is to manually specify pruning rates for each chunk. For example, below we show a simplified case of 4 embeddings with 2 chunks each. If we set the pruning rate to 50% for chunk 0 and 75% for chunk 1, then 2 chunks remain in the first chunk column, and 1 chunk remains in the second chunk column.
>
> | | chunk 0 utility | chunk 1 utility |
> |---|---|---|
> | Embedding 0 | 1.3 (keep) | 1.7 (keep) |
> | Embedding 1 | 0.9 (prune) | 1.6 (prune) |
> | Embedding 2 | 0.3 (prune) | 0.2 (prune) |
> | Embedding 3 | 2.1 (keep) | 1.5 (prune) |
> | Pruning rate | 50% | 75% |
> | Pruning threshold | <1.3 is pruned | <1.7 is pruned |
>
> If manual specification is used, then the pruning ratio for a chunk column remains fixed during training.
>
> The other way is to set a global pruning rate. For example, if a global pruning rate of 50% is used, then across all chunk columns, the 4 chunks with the highest utility values will remain, as shown below.
>
> | | chunk 0 utility | chunk 1 utility |
> |---|---|---|
> | Embedding 0 | 1.3 (prune) | 1.7 (keep) |
> | Embedding 1 | 0.9 (prune) | 1.6 (keep) |
> | Embedding 2 | 0.3 (prune) | 0.2 (prune) |
> | Embedding 3 | 2.1 (keep) | 1.5 (keep) |
> | Pruning rate (global) | 50% | 50% |
> | Pruning rate (chunk) | 75% | 25% |
> | Pruning threshold | <1.5 is pruned | <1.5 is pruned |
>
> If a global pruning rate is used, then the pruning ratio for a chunk column can fluctuate during training, because it is affected by utilities in other chunks.

---

> ### Author Response · Authors · 2023-11-16
> **Response to questions - Q3**
>
> **Q3**. In Section 3.3, you introduce the Virtually Hashed Physically Indexed (VHPI) embedding table design, which is a modification of AdaEmbed. Can you provide more details on how VHPI mitigates the issue of memory fragmentation and how it compares to other embedding table designs?
>
> Compared to a traditional 2D array design, the following example shows how FIITED solves the memory fragmentation issue. Consider a case of 4 embeddings, each having a different dimension with a maximum dimension of 8. Here we use numbers to indicate unpruned parts and _ for pruned parts.
>
> Embedding 0: 0000____
>
> Embedding 1: 111111__
>
> Embedding 2: 22_____
>
> Embedding 3: 33333333
>
> In the traditional design, the memory will look like: 0000____111111__22_____33333333. The free memory is scattered around in separate small segments, and it is not easy to utilize.
>
> In FIITED's design, only unpruned chunks are stored. In the VHPI hash table, addresses of the unpruned chunks are kept in order to read embedding chunks from the VHPI embedding table. For example, the VHPI hash table and the embedding table can look like this for the case above (with K=4, K is the number of chunks):
>
> --------
>
> **VHPI hash table**:（each entry has addresses of chunks, chunk utility values and a bit mask showing whether the chunks are pruned; invalid chunk addresses for pruned chunks are shown as N/A for clarity)
>
> Entry for Embedding 0:     (9, 5, N/A, N/A, chunk utility values, 1100)
>
> Entry for Embedding 1:     (4, 6, 1, N/A, chunk utility values, 1110)
>
> Entry for Embedding 2:     (0, N/A, N/A, chunk utility values, 1000)
>
> Entry for Embedding 3:     (2, 8, 3, 7, chunk utility values, 1111)
>
> --------
>
> **VHPI embedding table**:
>
> index 0,1 :     22,11
>
> index 2,3 :     33,33
>
> index 4,5 :     11,00
>
> index 6,7 :     11,33
>
> index 8,9 :     33,00
>
> --------
>
> For example, if we want to access embedding 0, we will first check its hash table entry, and see that there are 2 unpruned chunks at index 9 and 5 in the embedding table, and 2 pruned chunks. Then, we can get the embedding chunks from the VHPI embedding table by reading the values at index 9 and 5. The chunks are then concatenated to construct embedding 0.
>
> Now the memory looks like 22113333110011333300, with unpruned chunks stored together and free memory not in segments.
>
> Most recommender systems nowadays use the traditional 2D array storage structure. The VHPI storage system proposed by FIITED enables real memory saving during training.

---

> ### Author Response · Authors · 2023-11-17
> **Response to questions - Q4,5,6**
>
> **Q4**. In Section 4.2, you evaluate the performance of FIITED on four datasets, including three click-through rate prediction datasets and one classification dataset. Can you provide more details on the hyperparameters used in your experiments, such as the learning rate and batch size, and how they were selected?
>
> The learning rates and batch sizes were grid searched with some guidance from previous research papers that used the same datasets. We selected hyperparameters that yielded accuracy comparable to existing research papers using the same datasets. The details are provided below:
>
> | Dataset | Learning rate | Batch size |
> |---|---|---|
> | Criteo Kaggle | 1e-3 | 4096 |
> | Criteo Terabyte | 1e-3 | 2048 |
> | Avazu | 1e-3 | 4096 |
> | MovieLens-20M | 1e-3 | 512 |
>
> **Q5**. In Section 4.3, you compare the performance of FIITED with that of AdaEmbed, a state-of-the-art in-training embedding pruning method. Can you provide more details on how AdaEmbed works and how it differs from FIITED?
>
> AdaEmbed is a special case of FIITED where the number of chunks K is 1, i.e., every embedding is just 1 chunk. If this 1 chunk is pruned, then the embedding is pruned entirely. In AdaEmbed, an embedding vector is either totally unpruned or totally pruned.
>
> FIITED is a generalization of AdaEmbed and allows an embedding to be pruned in a fine-grained way.
>
> **Q6**. In Section 5, you discuss the limitations of your approach, including the trade-offs between memory savings and model quality, and the limitations in terms of scalability and computational complexity. Can you provide more details on how these limitations might impact the applicability of your approach to large-scale datasets?
>
> When it comes to scaling to large-scale datasets that requires a large number of embeddings, the model quality is limited by the amount of memory available if FIITED is used to fit the model into the on-device memory. If the amount of on-device memory is fixed, then there will be a point where the pruning rate needs to be so high that the model quality will suffer. Then, to ensure model quality, we will need to increase the amount of memory by, for example, increasing the number of devices used for training. A contribution of FIITED is that, given a memory budget, FIITED can figure out how to smartly prune embeddings to fit within the memory budget while achieving relatively good model quality compared to baselines in the field.
>
> As for the performance overhead of FIITED in terms of training time, it is a tug of war between the extra computation introduced by FIITED (as discussed in Q1) and the amount of computation and data transfer saved by FIITED. Computation is saved by increasing the sparsity in data. If the model is too big to fit in on-device memory, expensive data transfer is usually needed to obtain embeddings from off-device memory, and FIITED can save such data transfer by drastically reducing the model size.
>
> Please let us know if you have any questions or comments! Thank you.

---

### Official Review · Reviewer_CGz8 · 2023-11-01

**Soundness:** 4 excellent
**Presentation:** 4 excellent
**Contribution:** 4 excellent
**Rating:** 8
**Confidence:** 4

**Summary:**

In this paper, the authors have proposed a novel embedding dimension optimization method that adjusts the dimension of each individual embedding vector continuously during training; assigning longer dimension to more important embeddings while adapting to dynamic changes in data.

**Strengths:**

The authors are solving a very interesting and useful problem. Data storage efficiency is a very industrial problem and this effort is very relevant.

**Weaknesses:**

None

**Questions:**

None

---

> ### Author Response · Authors · 2023-11-15
> **Thank you for your review!**
>
> Thank you for your review! Please let us know if you have any questions or comments.

---

### Official Review · Reviewer_1xeD · 2023-11-03

**Soundness:** 2 fair
**Presentation:** 3 good
**Contribution:** 2 fair
**Rating:** 3
**Confidence:** 4

**Summary:**

The paper introduces FIITED, a novel embedding pruning technique tailored for very large-scale recommendation models. This method is rooted in the understanding that not all embedding dimensions are equally important, thus allowing for selective pruning. FIITED's design philosophy hinges on dimensional chunking, where embeddings are split into chunks by dimensions. The significance of these chunks is gauged through a utility-based approach, taking into account both the access frequency and the gradient values, to avoid prematurely evicting valuable chunks. The paper's design also proposes a system for addressing memory fragmentation issues: the system comprises a hash table for efficient embedding retrieval, an embedding table to store chunks, and a chunk address manager to maintain and manage memory allocation. Through this approach, FIITED aims to reduce the memory footprint of large-scale recommendation models without substantially compromising performance.

**Strengths:**

1. This paper presents a novel approach to embedding dimensionality reduction, innovatively combining dynamic chunk eviction strategies with utility considerations based on access frequency. The decision to merge small free memory chunks is a thoughtful addition, indicating a holistic approach to the problem.
2. The system's design, especially its implementation on both open-source and internal frameworks, suggests that FIITED is versatile and can be potentially adapted to various other architectures and platforms.
3. Tackling embedding dimensionality and memory efficiency is crucial in the domain of deep recommendation systems. The paper's contribution promises potential real-world impacts, particularly for large-scale applications.

**Weaknesses:**

1.	At the heart of FIITED is the utility-based approach to determine chunk significance. However, basing eviction decisions purely on utility scores might introduce biases. For instance, recent chunks might gain a temporary high utility, leading to potentially premature evictions of other valuable chunks.
2.	This approach does not consider the individual significance of dimensions within a chunk, leading to potential information loss.
3.	While the chunk address manager maintains a free address stack, this design assumes that the most recently evicted space is optimal for the next allocation. This might not always be the case, especially when considering the locality of data and frequent access patterns.
4.	The system heavily depends on the hash table to fetch and manage embeddings. This approach, while efficient in accessing chunks, might lead to hashing collisions even though the design ensures a low collision rate. Any collision, however rare, can introduce latency in access times or even potential overwrites.
5.	The methodology leans heavily on access frequency to decide on embedding significance. However, frequency doesn't always equate to importance. There could be rarely accessed but critically important embeddings, and the method might be prone to undervaluing them.

**Questions:**

1. In the context of FIITED, access frequency influences the utility. When the system frequently accesses certain recent chunks, those chunks might temporarily have a higher utility score compared to older chunks—even if those older chunks are historically more important. Although FIITED utilize the gradients of the chunk to mitigate such biases, The gradient might not always represent the "big picture" during the training process. For example, in the context of complex models, significant gradients might lead the optimization towards local minima rather than a more global optimal solution. Especially in the early stages of training, gradients can be noisy. This noise might mislead the system into thinking an embedding chunk is more or less important than it truly is. The authors should discuss more how it handles such potential fluctuations in utility values or if any smoothing mechanisms are in place.
2. Dimensions in embeddings often have dependencies, where certain dimensions complement or refine the information in others. Chunking might break these dependencies.  This can cause the model to lose subsets of interrelated features, compromising its understanding.Moreover, embedding' dimensions might not have uniform importance. By dividing them into chunks, there's a risk that a chunk might have a mix of crucial and less vital dimensions. If the less important dimensions in a chunk increase its eviction likelihood, it might inadvertently lead to evicting more essential dimensions too.The authors may consider analyzing the dimensional dependency and significance first, as well as the adaptive chunking as a fixed chunk size may not always be optimal.
3. The address manager simply re-allocates the most recently evicted space (due to the stack-based approach), it might re-allocate space that has high temporal locality. Thus, it could be evicting data that will be frequently accessed shortly after. This isn't efficient, as it could result in a high rate of data swapping in and out.Hence, simply using a stack to manage the evicted spaces could disrupt both forms of locality. If the system doesn't take into account the access patterns and just works on a "most recently freed" basis, it could lead to inefficiencies in memory access and potential performance bottlenecks, especially in scenarios where embeddings have strong temporal or spatial access patterns. Instead of purely stack-based address management, the authors may consider using a hybrid approach that takes into account both recent evictions and access frequency.
4. The system heavily depends on the hash table to fetch and manage embeddings. This approach, while efficient in accessing chunks, might lead to hashing collisions even though the design ensures a low collision rate. Any collision, however rare, can introduce latency in access times or even potential overwrites.The authors may need to employ a more robust hashing mechanism or a secondary probing mechanism to handle collisions more effectively.
5. Unlike whole embeddings that represent specific entities (like items), dimensions within embeddings capture specific features or facets of the represented entities. Some dimensions might be accessed less frequently because the specific feature they capture is less common in the dataset. Yet, this doesn't necessarily mean that feature is unimportant.For example, in e-commerce recommendation systems, there's often a "long tail" of products that aren't as popular as the mainstream ones but cater to specific needs or tastes. The dimensions representing these products might not be accessed as often, but they can be key to offering diverse and personalized recommendations to users looking for something unique. In addition, some users might have niche interests that aren't shared by the broader user base. The dimensions representing these niche interests might be accessed less frequently due to the smaller number of users with these interests. However, for those specific users, these dimensions are essential for generating accurate and satisfying recommendations.

---

> ### Author Response · Authors · 2023-11-17
> **Response to questions**
>
> Thank you for your review!
>
> **Q1**. How does FIITED handle potential fluctuations in utility values?
>
> FIITED takes history utility scores into consideration. As mentioned in Section 3.2, the utility value of a chunk in the i-th iteration is computed as follows:
>
> u(i) = γu(i − 1) + a(i)g(i)
>
> where u(i − 1) is the utility score in the previous iteration, a(i) is the number of accesses to the chunk in iteration i, g(i) is the L2 norm of the gradients computed for the chunk in iteration i, and γ ∈ (0, 1) is a decay parameter that controls the influence of the history utility value.
>
> We want to make sure small fluctuations do not affect the big picture, and at the same time adapt to the dynamic changes in data characteristics. For example, if certain embeddings were "hot" in the past but no longer relevant in the present, it may be better to give the space away to another embedding that is currently accessed frequently.
>
> As for the noisy gradients at the start of training, since it takes a while to fill up the embedding table before pruning needs to be triggered, the influence of noise is reduced.
>
> **Q2**. The authors may consider analyzing the dimensional dependency and significance first, as well as the adaptive chunking as a fixed chunk size may not always be optimal.
>
> This is a really interesting point and an interesting direction for future work. I think information loss is almost inevitable as far as pruning is concerned. Structured model pruning, like FIITED, risks losing potentially important model parameters but often offers more efficient storage and computation, compared with unstructured pruning. However, since embedding dimensions are symmetric in the arithmetic sense (the input dimensions are symmetric in dot product and fully connected layers used in MLP - note that embeddings considered in this work only correspond to 1 feature at a time and all dimensions within an embedding are always accessed together), there can potentially be a way to use additional regularization terms to help form certain group structures in the dimensions so that information loss is reduced.
>
> **Q3**. Instead of purely stack-based address management, the authors may consider using a hybrid approach that takes into account both recent evictions and access frequency.
>
> Optimizing the reallocation address management policy can definitely improve the spatial locality, but temporal locality-wise, it does not affect the performance much because pruning does not happen frequently (e.g., every few hundreds of iterations) in FIITED. When pruning does happen, it is a short, concentrated time period when a group of less important embedding chunks are discarded and re-initialized to make space for another group of more important chunks. There is 1 swap per pruned chunk, and the free address stack typically ends up empty at the end of the pruning and reallocation process. It is possible that certain chunk locations will keep getting swapped, but every 2 swaps are many training iterations apart.
>
> **Q4**. The authors may need to employ a more robust hashing mechanism or a secondary probing mechanism to handle collisions more effectively.
>
> The collision issue is not unique to FIITED, as hashing is commonly used in modern Deep Learning Recommender Models (DLRM) to cut down the large number of embeddings. Smart hashing techniques have been proposed to deal with the collision problem in DLRM. Currently, FIITED simply uses a hash table with a fairly large number of rows to make sure collision rarely happens, but moving forward, it can definitely benefit from an improved hashing scheme.
>
> **Q5**. Some dimensions might be accessed less frequently because the specific feature they capture is less common in the dataset. Yet, this doesn't necessarily mean that feature is unimportant.
>
> It is true that less accessed features may be important to a small group of users, but the problem of fair recommendation is a quite broad subject that will be hard to solve in one paper. Existing embedding pruning papers generally aim to optimize for model accuracy while reducing the model size as much as possible, which means the broader user base will inevitably be favored due to them contributing to the majority of validation/test samples. However, FIITED can potentially help to facilitate fairness. By saving memory usage during training, FIITED can allow the addition of more features into the model to diversify the representation of user / item traits. Moreover, one can design better utility metrics to promote fairness among features.
>
> Please let us know if you have any questions or comments! Thank you.

---

### Official Review · Reviewer_yjuq · 2023-11-03

**Soundness:** 3 good
**Presentation:** 3 good
**Contribution:** 2 fair
**Rating:** 5
**Confidence:** 3

**Summary:**

The paper proposes a novel method for optimizing the embedding dimension in to reduce the memory footprint during training. The proposed method  (FIITED), adjusts the dimension of each individual embedding vector continuously during training, assigning longer dimensions to more important embeddings while adapting to dynamic changes in data. The paper also proposes a novel embedding storage system based on virtually-hashed physically-indexed hash tables to efficiently implement the embedding dimension adjustment and effectively enable memory saving. Overall an interesting work with some shortfalls.

**Strengths:**

The paper presents a novel approach to optimizing the embedding dimension in DLRMs to reduce the memory footprint during training.
  he proposed method, FIITED, adjusts the dimension of each individual embedding vector continuously during training, which makes it more flexible and adaptable to dynamic changes in data.

The paper proposes a novel embedding storage system based on virtually-hashed physically-indexed hash tables to efficiently implement the embedding dimension adjustment and effectively enable memory saving.

**Weaknesses:**

The paper could benefit from more detailed explanations of the proposed embedding storage system and how it works, in particular details on the utility value computation or a systematic example.

The paper does not provide a comparison of their approach with other state-of-the-art methods in the field, in particular hashing and bloom filter-based methods have been shown to provide similar benefits.

**Questions:**

How does the methods compare to hashing and bloom methods.
Could you provide more details on the methodology and perhaps a concrete example?

---

> ### Author Response · Authors · 2023-11-15
> **Response to questions**
>
> Thank you for your review!
>
> **Q1**. How does the methods compare to hashing and bloom methods?
>
> If "hashing" here refers to methods that adjust the hash size of embedding tables (please let us know if "hashing methods" mean something else here), FIITED takes a different approach which adjusts the embedding dimension of embedding tables. The two approaches are related, as setting embedding dimensions to 0 is equivalent to reducing the hash size. But I think they also are, to some extent, orthogonal to each other and can be combined together, for example, we can adjust the embedding dimension after adjusting the hash size first.
>
> Similarly speaking, bloom filter-based methods also adjust the hash size of embedding tables. Compared with bloom filter-based methods, the unique advantage of FIITED is that FIITED exploits the fact that different embeddings do not need the same number of bits, which gives FIITED more opportunities to cut down the embedding size. But again, I think it is possible to combine these two types of methods, i.e., using bloom filter-based methods to cut down the number of rows first and then using FIITED to cut down the number of columns. It will be interesting to see to what extent the two approaches complement (or compete with) each other.
>
> **Q2**. Could you provide more details on the methodology and perhaps a concrete example?
>
> As mentioned in Section 3.2, the utility value of a chunk in the i-th iteration is computed as follows:
>
> u(i) = γu(i − 1) + a(i)g(i)
>
> where a(i) is the number of accesses to the chunk in iteration i, g(i) is the L2 norm of the gradients computed for the chunk in iteration i, u(i − 1) is the history utility value of the chunk computed in the previous iteration, and γ ∈ (0, 1) is a decay parameter that reduces the influence of the history utility value. The utility value is updated in this way in every iteration. The initial utility value is 0 at the start of training for every chunk.
>
> As for the embedding storage system, examples are given below.
>
> (1) Embedding access
>
> Let's say we want to access the j-th embedding for sparse feature i. Without the FIITED design, we will need to access the j-th row in the i-th embedding table. But with the VHPI design, all the embeddings across all sparse features are stored together in 1 big table (i.e., the VHPI embedding table), and the size of the big table is decided by the desired pruning ratio. Only unpruned chunks are stored, which gives FIITED the ability to truly cut down memory usage while dynamically adjusting the embedding dimension.
>
> To look for this embedding in the big table, index (i,j) will be hashed to obtain a new index i' in the VHPI hash table, and the i'-th entry of the VHPI hash table will be read. This entry stores addresses of all the chunks of the embedding as well as whether they have been pruned. All the unpruned chunks are read from the 1 big embedding table according to their addresses, and the pruned chunks are treated as all zeros. All the chunks are then concatenated to form the embedding we are looking for.
>
> (2) Embedding eviction and allocation
>
> Embedding eviction and allocation mainly happen during pruning. During pruning, the utility values of embedding chunks are sorted to decide a pruning ratio. For example, if a global pruning ratio of 75% is used, then for the following simplified example of 2 embeddings with 2 chunks each, only the chunk with the highest utility 1.5 is unpruned while the remaining 3 are pruned.
>
> | | chunk 0 utility | chunk 1 utility | VHPI hash table entry |
> |---|---|---|---|
> | Embedding 0 | 1.5 | 1.0 | [address of chunk 0, address of chunk 1, utility values (1.5, 1.0), bit mask: 10] |
> | Embedding 1 | 0.9 | 0.4 | [address of chunk 0, address of chunk 1, utility values (0.9, 0.4), bit mask: 00] |
>
> Whether the chunks have been pruned in an embedding is indicated in a bit mask in the embedding's VHPI hash table entry. For the case shown above, a 2-bit mask is used, with the 1st (2nd) bit showing whether chunk 0 (1) is present. If a chunk is present (i.e., unpruned), its address in the hash table entry is valid and shows where the chunk is located in the VHPI embedding table. If a chunk is not present (i.e., pruned), its address in the hash table entry is invalid and it is not stored in the VHPI embedding table.
>
> If a chunk goes from unpruned to pruned, it will be evicted. Its address in the VHPI embedding table will be passed to the free address stack, and its bit will be toggled to 0.
>
> If a chunk goes from pruned to unpruned, it will be allocated. Its bit will be toggled to 1, and it will get an address from the free address stack. The address indicates where the chunk is located in the VHPI embedding table.
>
> Please let us know if there are any questions or comments! Thank you.

---

### Meta-Review · Area_Chair_KM5v · 2023-12-01

**Metareview:**

Reviewers generally found the paper to provide a novel approach to optimising the embedding dimensionality in DLRMs. However, there were a large number of questions and concerns raised, primarily around the comparison, conceptually & empirically, against existing approaches. The author response gave some clarification on the former. However, the latter appears to remain unclear. Further, given the large number of changes that would be required to the manuscript to incorporate all the questions raised, it appears best for the paper to undergo another round of review.

**Justification For Why Not Higher Score:**

Four reviewers raised multiple concerns on the paper around conceptual and empirical comparison to existing approaches. Incorporating responses to all of these would result in a substantially different paper, which would be best served by a fresh round of review.

**Justification For Why Not Lower Score:**

N/A

---

### Decision · Program_Chairs · 2024-01-16

Reject